# Dependence of Temperature Rise on the Position of Catheters and Implants Power Sources Due to the Heat Transfer into the Blood Flow

Hossein Zangooei [1], Seyed Ali Mirbozorgi [1] and Seyedabdollah Mirbozorgi [2,*]

1   Mechanical Engineering Department, University of Birjand, Birjand 9717434765, Iran;
zangooei93@gmail.com (H.Z.); samirbozorgi@birjand.ac.ir (S.A.M.)
2   Electrical and Computer Engineering Department, University of Alabama at Birmingham (UAB),
Birmingham, AL 35294, USA
*   Correspondence: samir@uab.edu; Tel.: +1-205-934-8412

**Abstract:** This work provides a numerical analysis of heat transfer from medical devices such as catheters and implants to the blood flow by considering the relative position of such power sources to the vessel wall. We have used COMSOL Multiphysics® software to simulate the heat transfer in the blood flow, using the finite element method and Carreau—Yasuda fluid model (a non-Newtonian model for blood flow). The location of the power source is changed (from the center to near the wall) in the blood vessel with small steps, while the blood flow takes different velocities. The numerical simulations show that when the catheter/implant approaches the vessel wall, the temperature increases linearly for ~90% of the radial displacement from the centerline position to the vessel wall, while for the last 10% of the radial displacement, the temperature increases exponentially. As a result, the temperature is increased significantly, when changing the position of the catheter/implant from the centerline to the area adjacent to the vessel wall.

**Keywords:** implantable devices; catheters; heat transfer; blood flow; thermal analysis

## 1. Introduction

The internal temperature of the human body is almost 309.6 K. When the body is at rest, the normal range of this temperature is between 309.15 K to 311.15 K, while when doing heavy exercise, this temperature can temporarily increase to 314.15 K [1]. The human body has an internal mechanism for controlling temperature (cooling and heating) to return any abnormal temperature changes to the normal range [1]. This mechanism is mainly adapted for controlling external factors such as environment temperature variation [2–6]. Any changes in the local temperature of the body can be detected by the temperature control mechanism to maintain the safety of the body, while if it exceeds the normal range, it can damage the body tissues [7–12].

The use of catheters and implantable devices is increasing for diagnostic and therapeutic applications [13–16]. The catheters and implantable devices are widely used technologies for ultrasound imaging (Imaging Intra-Cardiac Echocardiography, ICE, and Intra-Vascular Ultrasound, IVUS), tethered and wireless implants such as brain implants, leadless heart implants, bladder implants, etc. [17–20]. These miniaturized electronics devices operate inside the body at different locations and consume electrical power [12,21,22]. The electrical power is consequently converted to heat, which is dissipated in the surrounding body tissues. This regional heat causes a temperature rise in the human body tissues which needs to be controlled and limited to ensure safety [8,10].

Figure 1 shows the conceptual schematic of the catheters and implantable devices for various applications at the different human body positions. Based on the study presented in [23], the local heat generated by the implants in the human body is significantly proportional to the blood velocity across the implants. On the other hand, the blood velocity is

not constant everywhere in a blood vessel (it reduces to zero velocity near the vessel wall). Therefore, the position of the implant in the blood vessel is an essential parameter for safety considerations and adds extra limits to the design by reducing the maximum allowable power that the implants can consume [16,23].

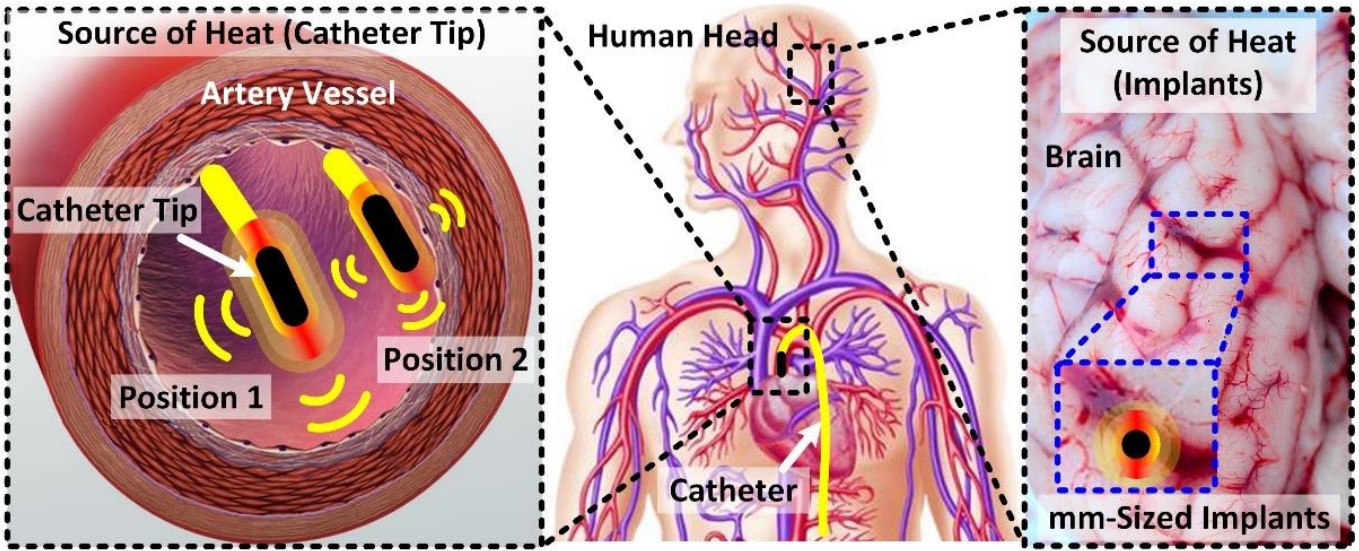

**Figure 1.** The conceptual schematic of the positions of the catheters and implants in the human body (located in/near the blood vessel).

In this work, we have changed the location of implants in a blood vessel model (from the center to near the wall) with small steps, using COMSOL Multiphysics® software (version 5.6, Stockholm, Sweden) and evaluated the local temperature rise in the human body. A comprehensive analysis of heat transfer is done by comparing the temperature rises at different implants positions. As a result, we have modified the temperature rise prediction equation presented in [23] by adding the new parameter of the implant's position into it. Section 2 presents an overview of the design and system modeling method. Section 3 presents the simulation results and is followed by a discussion in Section 4. Finally, Section 5 presents the conclusion.

## 2. System Modeling

We have modeled a 3-dimensional Blood-Vessel Implant (BVI) system based on [23], in which the catheter/implant was located at the centerline of the vessel ideally for all investigated different scenarios. In this work, we have changed the location of the catheter/implant from the centerline to the area adjacent to the vessel wall with small steps. The cross-section schematic (2D, top, and side) of the modeled BVI system is shown in Figure 2 for two positions of the implant (centerline, Figure 2a, and near the vessel wall, Figure 2b). Figure 2a,b indicate (1) the inlet and outlet blood flow, (2) the catheter and its tip (implant/probe), (3) the vessel and sinusoidal blood flow at the inlet, and (4) the dimensions of the model. $L_1$, $L_2$, and $L_3$ are the lengths of the modeled vessel, implant/probe, and catheter, respectively. $D_1$ is the inner diameter of the vessel and $D_2$ is the outer diameter of the catheter/implant. In addition, Xcp determines the location of the catheter proportion to the centerline of the blood vessel. The parameters of the dimensions of the BVI model are summarized in Table 1.

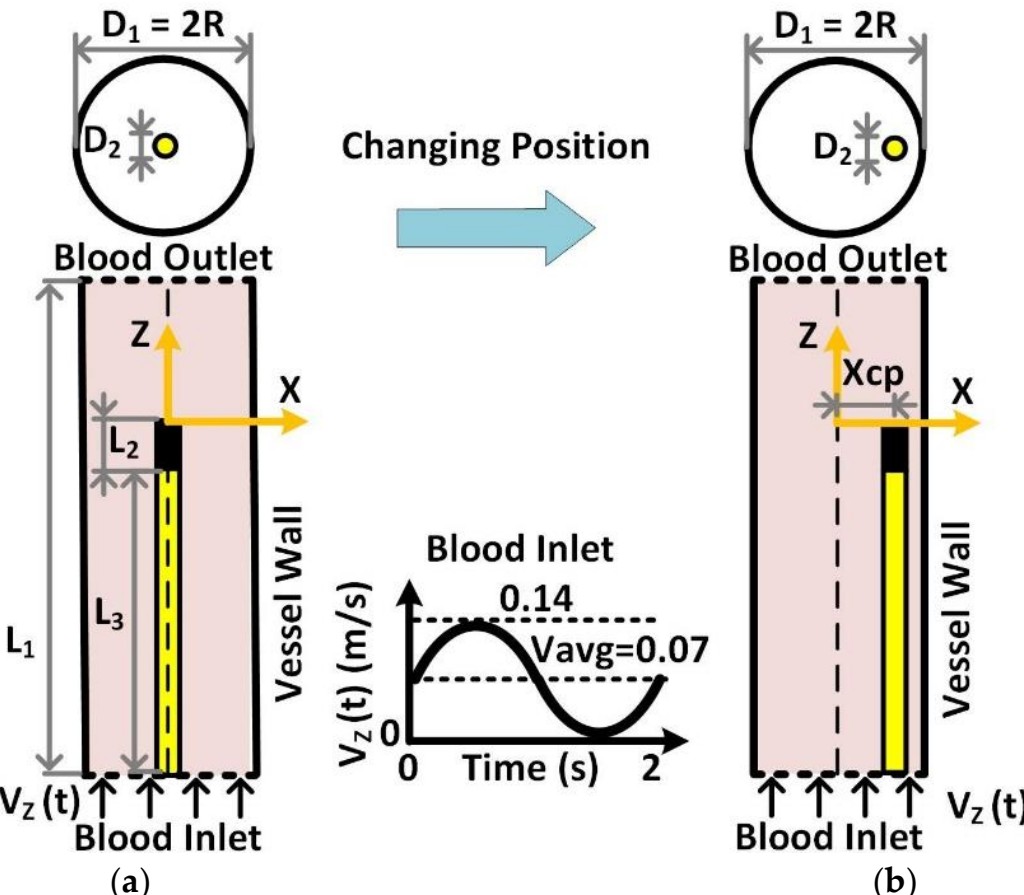

**Figure 2.** Conceptual schematic of the BVI system, including the blood vessel and catheter-probe/implant, (**a**) probe/implant in the centerline of the vessel (Xcp = 0 mm), and (**b**) probe/implant near the vessel wall (Xcp = 6 mm). Inset: the applied blood flow velocity profile as a function of time at the inlet of the blood vessel.

**Table 1.** The parameters of the dimension of the BVI model.

| Parameters | Value (mm) | Description |
|:---:|:---:|:---:|
| D1 | 13 | Diameter of the Blood Vessel |
| D2 | 0.8 | Diameter of the Catheter |
| Xcp | 0–6 | Position from Center |
| L1 | 80 | Length of the Blood Vessel |
| L2 | 5 | Length of the Catheter Tip, Probe/Implant |
| L3 | 50 | Length of the Catheter |

We have used the set of Navier–Stokes (N-S) equations (including continuity (1), momentum (2), and energy (3) equations) and a non-Newtonian fluid model (Carreau–Yasuda, (4)) for the proposed model [14–16].

$$\frac{\partial \rho}{\partial t} + \nabla \cdot \left( \rho \vec{V} \right) = 0, \tag{1}$$

$$\rho \left( \frac{\partial \vec{V}}{\partial t} + \left( \vec{V} \cdot \nabla \right) \vec{V} \right) = -\nabla p + \nabla \cdot \tau_{ij} + f, \tag{2}$$

$$\rho c_p \left( \frac{\partial T}{\partial t} + \vec{V} \cdot \nabla T \right) = -\nabla \cdot q + Q, \tag{3}$$

$$\frac{\mu - \mu_\infty}{\mu_0 - \mu_\infty} = \left[ 1 + (\lambda \dot{\gamma})^\alpha \right]^{(n-1)/\alpha}, \tag{4}$$

where in N-S Equations (1)–(3), $\vec{V} = V_x i + V_y j + V_z k$ is the fluid flow velocity vector, $p$ is the pressure, $\rho$ is the fluid density, $\tau_{ij} = \mu \dot{\gamma}$ is the shear stress tensor of blood (as an incompressible flow, where $\mu$ is the dynamic viscosity and $\dot{\gamma}$ is the shear rate), $f$ is the volumetric body force (if existing), $c_p$ is the specific heat capacity at constant pressure, $T$ is the temperature, and $Q$ is the heat source. In addition, to express Fourier's Law of Heat Conduction, $q$ is the heat flux in $q = -k\nabla T$, where $k$ is the thermal conductivity of the fluid (blood). In the Carreau–Yasuda non-Newtonian fluid model, (4), $\mu_\infty$, $\mu_0$, and $\lambda$ are the viscosity at an infinite shear rate, the zero-shear-rate viscosity, and the relaxation time (seconds), respectively. The parameters $\alpha$ and $n$ control the blood dilatation behavior due to shear stress [23].

The velocity of the blood flow in the human body is a periodic velocity, which can be modeled by a sinusoidal function, ideally [23,24]. We have generated and applied the periodic blood flow velocity at the input of the vessel model (Figure 2: Blood Inlet, Figure 2: Inset, periodic function, $V_Z(t)$). The modeled blood flow velocity function at the vessel inlet is presented in (5):

$$V_Z(t) = V_{avg}[1 + \sin(\pi t)], \tag{5}$$

where $V_{avg}$ is the blood flow average velocity at the input of the vessel [24]. The BVI system is modeled using COMSOL Multiphysics®, and this software has solved the governing Equations (1)–(3), using the finite element method. We have used the previously developed boundary conditions presented in [23] for the proposed BVI model in this work.

## 3. Simulation Results

We have used the modeled BVI system to evaluate the temperature rise in the blood due to the heat generated by the implant in the blood vessel. We have changed the position of probe/implant from the centerline of the blood vessel (Xcp = 0 mm) to near the vessel wall (Xcp = 6 mm) in four steps in the BVI model and compared the temperature rises. In Figure 3, the temperature contours around the implant are illustrated for two positions of Xcp = 0 mm and Xcp = 5 mm at $t$ = 1.5 s, while the heat flux at the implant/probe surface was set to be 15,000 W/m². This figure includes the results at the different average velocities of blood flow such as 0.1 m/s, 0.6 m/s, and 1.4 m/s. Based on these simulation results, the implant has experienced higher levels of temperature rises of 2.5 K, 0.91 K, and 0.81 K when it is located near the vessel wall (Xcp = 5 mm), for blood velocities of 0.1 m/s, 0.6 m/s, and 1.4 m/s, respectively. As it was expected, (1) when the implant is close to the wall, the BVI system experiences a higher temperature rise, and (2) a higher blood flow velocity causes a lower temperature rise.

Figure 4 shows the temperature rises in the blood vessel for different positions of the implant (Xcp = 0, 2.5, 5, and 6 mm). In this simulation, the heat flux and the average velocity of blood flow were set to be 15,000 W/m² and 1.4 m/s, respectively. As shown in Figure 4, for Xcp = 6 mm, the temperature is raised significantly at the gap between the implant and vessel wall due to the lower velocity of blood flow with the non-slip boundary condition. Therefore, it is critical to take the position of the implant as a vital parameter into account/assumption to ensure the safety of using an implant/catheter. Figure 5 indicates the numerical results of the temperature rise over a line at the tip of the implant (at Z = 55 mm in Figure 4) as a function of the blood vessel diameter (X: from −2.5 mm to +6.5 mm). In this simulation, the implant is located in different positions of Xcp = 0, 2.5, 5, and 6 mm.

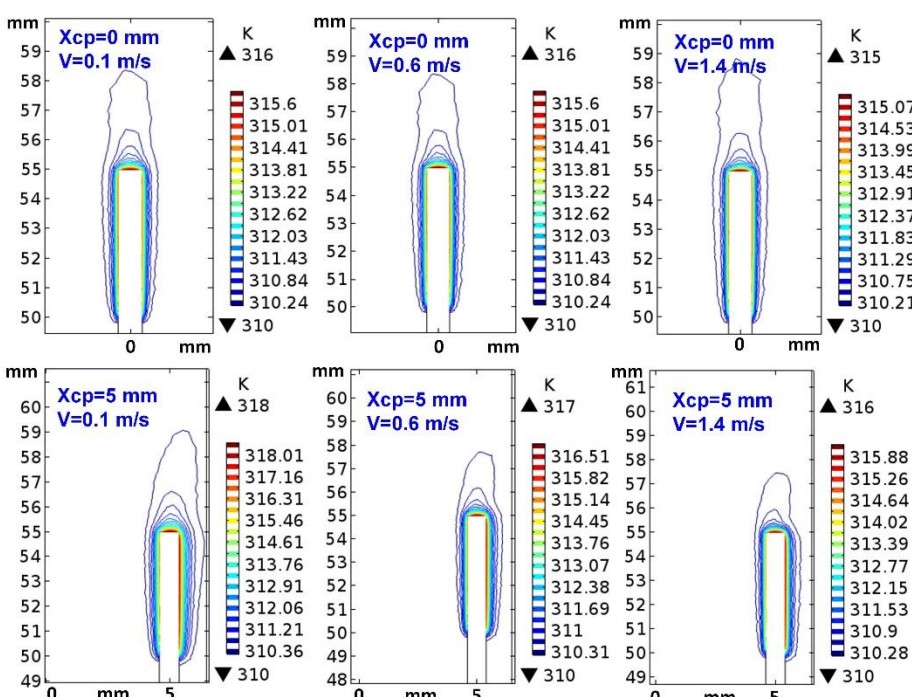

**Figure 3.** The simulated temperature contours while the average velocities are 0.1, 0.6, and 1.4 m/s (with Xcp of 0 and 5 mm and heat flux of 15,000 W/m²).

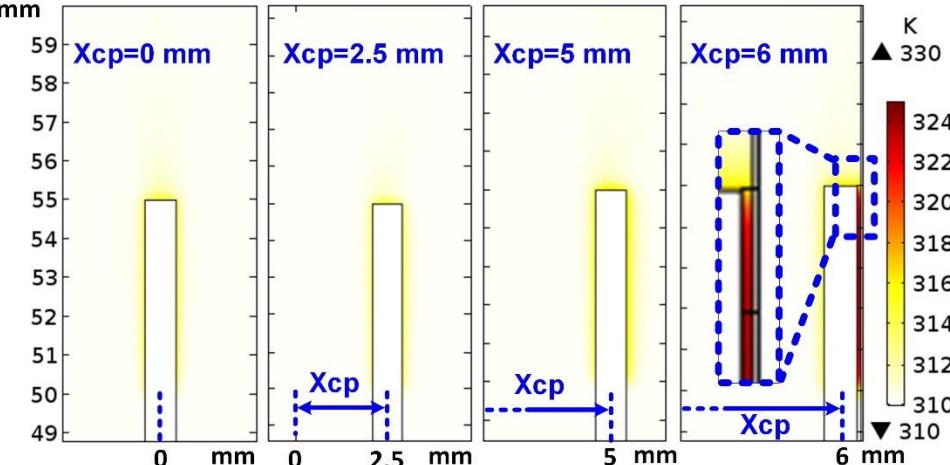

**Figure 4.** The simulated temperature contours while the implant is located at Xcp of 0, 2.5, 5, and 6 mm (with the average velocity of 1.4 m/s and heat flux of 15,000 W/m²).

Figure 5 shows the temperature contours at the diametrical cross-section (top view) of the blood vessel and implant. The results indicate a nonlinear temperature rise from Xcp = 0 mm to Xcp = 6 mm, in which the difference between Xcp = 0 mm to Xcp = 6 mm on the tip of implant is ~2 K.

The temperature is raised exponentially near Xcp = 6 mm. Although the temperature rise at the tip of the implant is up to 317.6 K (Figure 5), the maximum temperature rise (up to 324 K) occurs at the gap between the implant and the vessel wall, as shown in Figure 4 (for Xcp = 6 mm). The temperature rise in the gap has exceeded the safe threshold of 315 K by 9 K. Figure 6 indicates the simulation results of the temperature changes as a function of heat flux (0–15,000 W/m²) in two positions of Xcp = 0 mm and Xcp = 5 mm, and three average velocities of 0.1 m/s, 0.6 m/s, and 1.4 m/s. These results show a linear relationship between temperature rise and heat flux rise. This linear relation is also obtained at the different blood flow velocities and the different positions of the implant. The temperature

difference for two positions of Xcp = 5 mm and Xcp = 0 mm and the blood flow velocities 0.1 m/s, 0.6 m/s, and 1.4 m/s are almost 1.15 K, 1.56 K, and 1.45 K, respectively (at the heat flux 15,000 W/m$^2$).

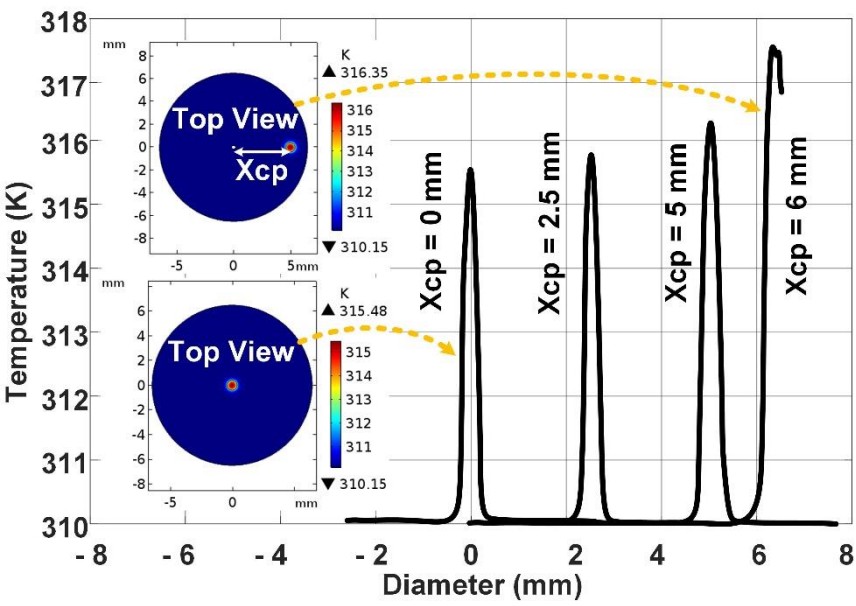

**Figure 5.** The numerical results of blood temperature as a function of the vessel diameter for implant heat flux of 15,000 W/m$^2$, average blood velocity of 1.4 m/s, and different locations of the implants (Xcp = 0, 2.5, 5, and 6 mm). The insets show the temperature contours at the cross-section of the vessel (at the implant tip).

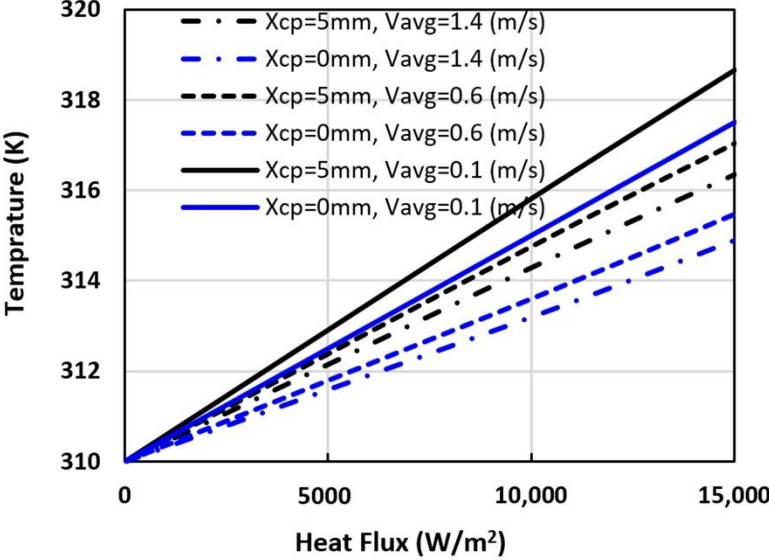

**Figure 6.** The numerical results of blood temperature rise as a function of the heat flux for two positions of Xcp = 0 mm and Xcp = 5 mm, and the average blood flow velocities of 0.1 m/s, 0.6 m/s, and 1.4 m/s.

## 4. Design Rule and Discussion

In this section, we have plotted the obtained results in a 3D fashion to visualize the relationships between the temperature rise, blood flow velocity, heat flux, and positions of an implant. We have also developed a correlation formula, corresponding to these results to present a design rule for safety check and estimating/predicting the temperature rise in the human body when using/designing an implant/catheter that produces the heat.

Figure 7a indicates a 3D visualization of simulated results where the temperature rise has been illustrated as a function of the average blood flow velocity and the heat flux for two positions of the implant (Xcp = 0 mm and Xcp = 6 mm). These results include the blood flow velocities up to 2 m/s and heat fluxes up to 100,000 W/m² to cover the heat flux ranges of ablation applications. This 3D plot shows two different levels of temperature rises for the implant positions of Xcp = 0 mm and Xcp = 6 mm, in which for the position Xcp = 6 mm, the temperature rise slope is higher than position Xcp = 0 mm. Figure 7b includes two top views of BVI system (cross-section at top of the implant) for the indicated implant positions. In addition, Figure 7b (at the top) shows a 2D temperature rise as a function of the average velocity of blood flow at the heat flux of 100,000 W/m². These 2D curves show that the temperature changes exponentially for both positions of the implant.

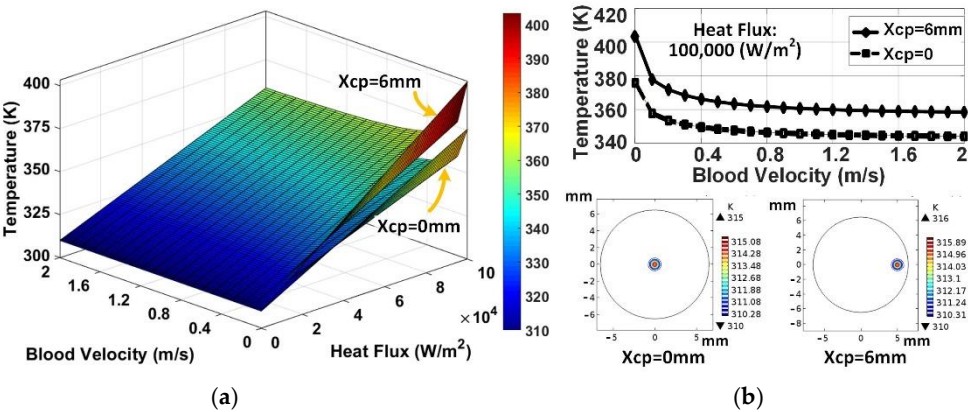

(**a**)                (**b**)

**Figure 7.** (**a**) The numerical results of blood temperature rise as a function of the blood average velocity and heat flux for two positions of the implant in the blood vessel, and (**b**) the temperature as a function of velocity for heat flux of 100,000 W/m² (at the **top**) and two positions of the implant.

According to the previous study and the new simulation results in this work, we have modified the temperature rise correlation formula in [14] by including the parameter of implant/catheter position in the blood vessel. This new correlation formula, (6), can be used to predict and estimate the temperature rise (K) in the human body accurately:

$$T = 310 + \frac{H(1 + 0.07\text{Xcp})}{3000}\left(1 + e^{-\sqrt{7V}}\right),\tag{6}$$

where Xcp (mm) is the distance between implant and centerline of the blood vessel, $V$ (m/s) is the blood flow average velocity, and $H$ (W/m²) is the heat flux produced by the implant. Based on (6), we have proposed a flowchart for the safety check design rule to estimate the temperature rise in the human body, shown in Figure 8.

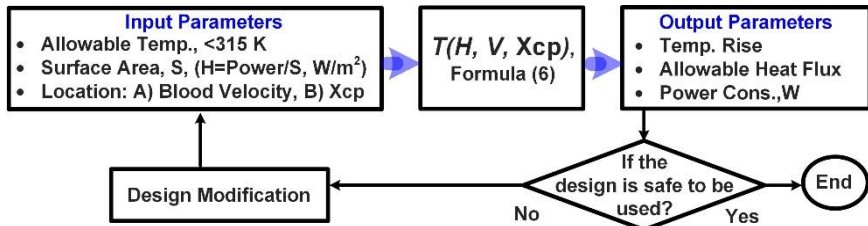

**Figure 8.** The proposed safety check flowchart for estimating and predicting the temperature rise during the process of designing implantable/catheter devices.

To calculate the heat flux produced by implants, it is required to know the electrical power consumption of the design and its surface area ($H$ = Power/Surface (W/m²)). The designers can use the flowchart to calculate the allowable heat flux that the system

can tolerate based on the location of the implant, blood velocity at that location, power consumption of the design, and its surface area.

## 5. Conclusions

A numerical simulation of the heat transfer from medical devices to the blood flow in the human body has been presented previously, while the devices were located at the centerline of blood vessels. However, in the present study, we have extended the results by including the vital parameter of implant location in different positions in the blood vessel. The proposed BVI system is modeled and simulated by COMSOL Multiphysics (using a Carreau–Yasuda Non-Newtonian model). We have correlated the simulation results into a formula to estimate/predict the temperature rise due to electrical power consumption of implant/catheter in the human body. This formula can be used to predict the temperature rise in blood vessel and check/evaluate the safety level of the design. The results show a significant dependency of temperature rise on the location of the implant at any level of heat flux and blood velocity. The temperature rise in the gap between the implant and the vessel wall at Xcp = 6 mm has exceeded the safe threshold of 315 K by 9 K.

**Author Contributions:** Project administration, S.A.M.; Supervision, S.M.; Writing—original draft, H.Z. All authors have read and agreed to the published version of the manuscript.

**Funding:** This research received no external funding.

**Institutional Review Board Statement:** Not applicable.

**Informed Consent Statement:** Not applicable.

**Data Availability Statement:** Not applicable.

**Conflicts of Interest:** The authors declare no conflict of interest.

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
