# Peer review of "Dependence of Temperature Rise on the Position of Catheters and Implants Power Sources Due to the Heat Transfer into the Blood Flow"

_electronics, doi:10.3390/electronics11121878_

Round 1

Reviewer 1 Report

This study by Hossein Zangooei et al reports a numerical simulation of the heat transfer from a medical devices to the blood flow in the human body. The manuscript is a continuation of a previous reported study where the heat transfer was only calculated when the catheter is located at a central position in blood vessel. In the present study, the authors extended the results by including the implant location in different positions in the blood vessel.  The major claim is that the position of the implant on the blood vessel has an influence on the temperature rise of the surrounding tissues.  It is reported that the temperature may exceed the safe threshold by 9º C

I cannot judge on the COMSOL simulations presented. They sound correct.  Concerning to the data presentation, I think Fig. 7 needs to be modified or rearranged. The 3D plot is very confusing and I cannot understand the information that this plot intends to show to the reader. Before this manuscript is accepted for publication in Electronics, I strongly recommend to reshape the 3D plot, in Fig.7  and include less information on it.

Reviewer 2 Report

This manuscript provides a numerical analysis of heat transfer from medical devices such as catheters and implants to the blood flow by considering the relative position of such power sources to the vessel wall. However, the pictures, formula as the design rule and references are hard to accept. I suggest major revision before further decision of acceptance or reject. The detailed comments are list as below:

(1) “while if it exceeds the normal range, it can damage the body tissues [1]-[12].”; “The use of catheters and implantable devices are increasing for diagnostic and therapeutic applications [1]-[14].”; “leadless heart implants, bladder implants, etc., [16]-[21].” Please mask sure if the reference number is right. I think it might be [10]-[12] and [13], [14]. Besides, where is Ref. [15]? Please double check carefully.

(2) In the abstract, “As a result, the temperature is increased by ~9 K, when ......” I suggest not indicate the absolute value here, because it is determined by the specific simulation model and boundary condition setting.

(3) Fig 1 is poor in quality, including different font sizes, text directions and low resolution.

(4) Parameters are unnecessary to list in Fig 2. Besides, please unify the font sizes and text directions.

(5) Time=1.5 s and H=15000 W/m2 are same for all subfigures in Fig 3, thus it is unnecessary to emphasize in each subfigure. Besides, the drawing format is not standard for a scientific paper. Same question in Fig 4, and the poor resolution and different font sizes should be paid attention.

(6) Parameters T=1.5 s, H=15,000 W/m2, V=1.4 m/s are unnecessary to appear in Fig 5.

(7) The layout is a mess in Fig 7, with subfigures obscuring each other. Please redraw it.

(8) I do not think Fig 8 is necessary. The equation T has already listed as formula (6).

(9) It is easy to know if the implant is far from the vessel wall, the heat will transfer quickly via the flowing blood with lower temperature rise. Hence, what is the necessity of this research, even if you get the result of 9K? What is the next or solution to avoid unsafe heat transfer? How can you avoid the inevitably close from the implant to the vessel wall?

(10) Line 56, “In this work, we have swept the location of implants in a blood vessel model”, how can you sweep the location? What tool do you use? What are the specific type, brand, and model parameters of the implant?

(11) Line 189, “we have modified the temperature rise correlation formula in [14] by including the parameter of implant/catheter position in the blood vessel.” The authors proposed the formula (6) as the safety check design rule via modification from [14]. However, how can you prove this formula is correct? It is not rigorous at all and hard to approve.

Round 2

Reviewer 2 Report

In Q4, “We had to keep the parameters list there since we don’t have any table in the paper to include the parameters and used values.” You can and need to add a table! List parameters in pictures is not professional for literatures! Please read papers in Nature or Science, almost no one will add parameters in pictures!

In Q9, “This work is the first research that has evaluated the level of temperature rise near walls.” Your statement is too conceited! How can you prove it? What’s more, you do not solve the problem with unsatisfactory arguing!

In Q10, “By using the word “swept”, we meant that we had changed the location of the implant from the center of the blood vessel to the wall with 1 or 2.5 mm sized steps.” However, I think you should change this word to avoid misunderstanding.

In Q11, Although the simulation results are accurate, we will need experimental evidence to prove it for future works.How can you prove the simulation is accurate? Any simulation that is not based on experiments is not credible. Your statement is too conceited.

Round 3

Reviewer 2 Report

Agree to accept for publication in this version.